# Non-invasive Ventilation Interventions for Skin Injury Prevention: Scoping Review

**Rita Azevedo** [1], **Tânia Manuel** [1] and **Paulo Alves** [2,*]

1   Centre for Interdisciplinary Research in Health (CIIS)–Wounds Research Lab, Universidade Católica Portuguesa, 4169-005 Porto, Portugal; s-arrazevedo@ucp.pt (R.A.); s-tmscarvalho@ucp.pt (T.M.)
2   Centre for Interdisciplinary Research in Health (CIIS)–Wounds Research Lab, Faculdade de Ciências da Saúde e Enfermagem, Universidade Católica Portuguesa, 4169-005 Porto, Portugal
*   Correspondence: pjalves@ucp.pt

**Abstract:** Background: Pressure ulcers associated with the non-invasive ventilation mask can significantly reduce the quality of life of the patient who needs this therapy. This study aims to identify clinical interventions to prevent skin lesions associated with the use of non-invasive ventilation medical devices. Methods: The Scoping Review followed the methodology of the Joanna Briggs Institute. For this study the research was carried out, during the month of January 2022, in several databases, such as PubMed, Web of Science, Scopus, EBSCOhost, RCAAP and OpenGrey, and studies published between 2010 and 2022 were included. Results: Of the 33 articles identified, 11 articles were included in this review, in which we identified several interventions for the prevention of skin lesions associated with the use of medical devices for non-invasive ventilation. The interventions identified include skin assessment, optimal fixation of the device, and the use of interfaces, namely, hydrocolloid or foam dressing under the NIV mask, among others Conclusion: This scoping review demonstrates that there is some scientific evidence for prevention, however the methodological approaches are very different, which makes it difficult to clearly describe the referenced interventions. This study was not registered.

**Keywords:** non-invasive ventilation; pressure ulcer; mask; equipment and supplies; nursing care

## 1. Introduction

The concept of Pressure Ulcer (PU) has undergone several changes along with the evolution of its investigation, and currently the PU is defined as a lesion that is located on the skin or in the underlying tissue as a result of the pressure, or the combination of the pressure with the torsion budgets, usually on a bony prominence [1]. About 95% of PUs are estimated to be preventable through the early assessment of the degree of risk and the adoption of appropriate measures in relation to them [2]. The treatment of PUs usually entails high costs, and in the United Kingdom this is estimated to vary between £1214 and £14,108 per year, representing a significant burden on the healthcare system, and the increase in cost varies with the severity of the ulcer, since healing time can be longer, as well as the incidence of complications [3].

The development of PU is associated with a series of determining factors, such as age, length of hospital stay, male gender, low albumin values and administration of vasopressors [4].

Medical Device-Related Pressure Ulcers (MDRPUs) develop when the underlying skin and/or tissue are subjected, on the part of the medical device, to sustained pressure, or even to shear forces [1]. The prevalence of MDRPUs is estimated at 10% and the incidence of the same lesions at 12% [5].

MDRPUs increase the risk of infection and sepsis, therefore increasing the risk to life for the patient [6]. These lesions can cause pain and alteration of the corporal image, since they can leave visible scars, decreasing the patient's quality of life and increasing the

length of hospital stay [6]. MDRPUs are demanding lesions at the level of treatment, since the source causing the pressure is the medical device, which will continue to be a factor, and in most situations cannot be removed, so its treatment is as difficult as the patient's dependence on the device [7]. The more extensive the use of medical devices, the greater the risk of developing MDRPUs [6].

The medical devices that are most associated with the development of MDRPUs are endotracheal tubes, oxygen delivery tubes, masks, intravenous catheters, splints and cervical collars [6]. The most common anatomical locations are the face and ears, the calcaneus and the leg, however these lesions can occur in any anatomical location that is under pressure from a medical device [6]. The treatment of MDRPUs is difficult as patients are dependent on the device and it is often not possible to remove it and relieve the pressure [7]. It is estimated that many of the medical devices commonly used in healthcare units, such as endotracheal tubes, naso-gastric tubes, oxygen tubes, non-invasive ventilation masks, urinary catheters and cervical collars, have undergone few changes in their configuration in recent decades. This could explain why their interface with vulnerable skin causes injuries associated with the pressure of the device [6].

The use of medical devices, namely devices that ensure effective non-invasive ventilation, is widespread in hospitals. Non-invasive ventilation (NIV) has gained significant attention as an alternative to invasive mechanical ventilation. While NIV is primarily associated with reduced complications, such as nosocomial infections and ventilator-associated pneumonia, one under-discussed but prevalent issue is the skin injury caused by the interfaces used in NIV [8]. This scoping review delves into the various interventions and practices used to mitigate skin injuries in patients receiving NIV.

With regards to PU associated with the non-invasive ventilation mask (NIV), the literature reports an incidence close to 20% [8,9]. The most frequently reported site of appearance of these lesions is the nasal bridge [9–11]. These lesions caused by the non-invasive ventilation mask can significantly reduce the patient's quality of life [10,12]. Inadequate adjustment of the mask can lead to the development of PUs, however, sometimes, even with the appropriate mask well adjusted, lesions may arise [13]. Strategies, such as the ideal adjustment of the mask, as well as hydration of the skin, are important factors in maintaining the integrity of the tissues during the use of NIV [14].

The development of a pressure ulcer associated with the NIV mask occurs in most patients within the first 48 h of treatment [11]. The pharmaceutical industry is working to improve NIV interfaces, however skin lesions often continue to develop. The use of dressing material under NIV masks appears to be a strategy, cited in the literature, that allows reduction in the incidence of these injuries, without interfering with ventilation. These lesions can be fatal, since they can prevent the maintenance of the application of non-invasive ventilation to a patient, thus aggravating the respiratory disorder [10,12]. Some studies indicate interventions that can be performed to reduce the risk of PUs associated with NIV masks, such as the application of hyper-oxygenated fatty acids [15] and the use of protective dressings under the mask [16,17].

Despite the growing development of interventions aimed at the prevention of MDR-PUs, there is no standardized application for their prevention in practice. Thus, the objective of this review is to identify interventions capable of preventing PUs associated with non-invasive ventilation masks in adults.

## 2. Materials and Methods

The performance of this Scoping Review followed the methodology of the Joanna Briggs Institute [18] and the guidelines established as the PRISMA (Preferred Reporting Items for Systematic Reviews and Meta-Analyses) model [19]. The sub-topics are the population to be included in the research, the formulation of the research question, the eligibility criteria, the strategy, and the databases used to obtain the relevant information.

The definition of the starting question followed the strategic parameters P (Population), C (Concept), C (Context), and the search is guided by the following question:

"What are the interventions needed to prevent pressure ulcers associated with the non-invasive ventilation mask in adults?"

*2.1. Inclusion Criteria*

This scoping review considered studies that included adult participants, i.e., aged 18 years or older, with no age limit, regardless of gender. The health condition, as well as the origin of the participant (home or hospitalization) was not a factor of exclusion, since we intend to different interventions that prevent these injuries, regardless of the context in which they occur. We focused on studies that explored the interventions that can be performed to prevent pressure ulcers associated with the medical device—non-invasive ventilation mask—in adults. Qualitative studies, quantitative and mixed, as well as randomized clinical trials, prospective studies, retrospectives, cohorts, case studies, descriptive studies and systematic reviews, were considered. Additionally, protocols and guidelines were also included. This review had no limit in the context, and thus studies from home and community contexts, hospital contexts or research contexts were considered.

*2.2. Research Strategy*

The research strategy aimed to find both published and unpublished primary studies, systematic reviews, case studies, protocols, and guidelines.

The search was performed in several databases, such as PubMed, Web of Science, Scopus, EBSCOhost, RCAAP and OpenGrey, as shown in Table 1, in January 2022. The same search terms/descriptors were used to carry out the search in the different databases: EBSCOhost, Pubmed, Scopus, Web of Science and RCAAP. After an initial reading of articles on the topic, the descriptors and synonymous terms were employed, as well as expressions found in the articles, in order to include as many documents as possible that could enrich the research. As shown in Table 1, the terms/descriptors are: Query 1: "pressure ulcer" OR "pressure injury"; Query 2: "Intervention OR", "Preventive Intervention" OR "Prevention"; Query 3: "medical devices" OR "Medical devices-related pressure ulcer"; Query 4: "non-invasive ventilation" OR "Ventilation Masks" OR "Non-invasive ventilation devices" and, finally, combinations of the same. Studies with access to full text in English, Portuguese and Spanish, published from 2010 to the present year, were considered for inclusion. The languages.in which the reviewers were most fluent were selected to ensure the correct interpretation of the studies included in this review. As English is considered a universal language, it was deemed adequate in order to cover this topic broadly. The research of guidelines and protocols were carried out in relation to the associations that work in this field, namely EPUAP, EWMA, NPIAP, ELCOS and APTFeridas. The research was carried out in recognized associations that work in the area of wounds, as well as in OpenGrey. It was carried out manually regarding the search for documents, protocols and guidelines that could bring relevant contributions to this review, in the absence of a search engine that would allow us to carry out the search with the defined descriptors. This search was exhaustive, in order to find these documents through the tabs and computer libraries available on the website of the previously mentioned associations. Taking into account the inclusion criteria, articles were selected by two independent reviewers, according to titles and abstracts.

After the research was re-developed, the studies were grouped together on the Mendeley Desktop® version 1.19.8 and those that were duplicates were automatically removed.

Figure 1 presents the PRISMA diagram, in which the articles initially identified, the entire selection period and the 11 final articles included in this scoping review are verified.

**Table 1.** Database search.

| Search | Query | EBSCOhost | PubMed | Scopus | Web of science | RCAAP | OpenGrey | EPUAP | EWMA | NPIAP | ELCOS | APTFeridas |
|---|---|---|---|---|---|---|---|---|---|---|---|---|
| #1 | "Pressure ulcer" OR "pressure injury" | 32,895 | 6517 | 33,931 | 5789 | 47 | | | | | | |
| #2 | "Intervention" OR "Preventive Intervention" OR "Prevention" | 4,793,500 | 588,9610 | 7,320,992 | 2,154,508 | 5371 | | | | | | |
| #3 | "Medical devices" OR "Medical devices-related pressure ulcer" | 54,607 | 43,246 | 203,976 | 26,916 | 0 | | | | | | |
| #4 | "Non-invasive ventilation" OR "Ventilation Masks" OR "Non-invasive ventilation devices" | 9517 | 8724 | 22,454 | 6382 | 0 | | | | | | |
| | #1 AND #2 AND #3 AND #4 | 7 | 29 | 97 | 6 | 0 | | | | | | |
| Limited to articles in the English, Portuguese and Spanish language published between 2010 and 2022 | | 7 | 29 | 92 | 6 | 0 | 0 | 1 | 0 | 0 | 0 | 0 |

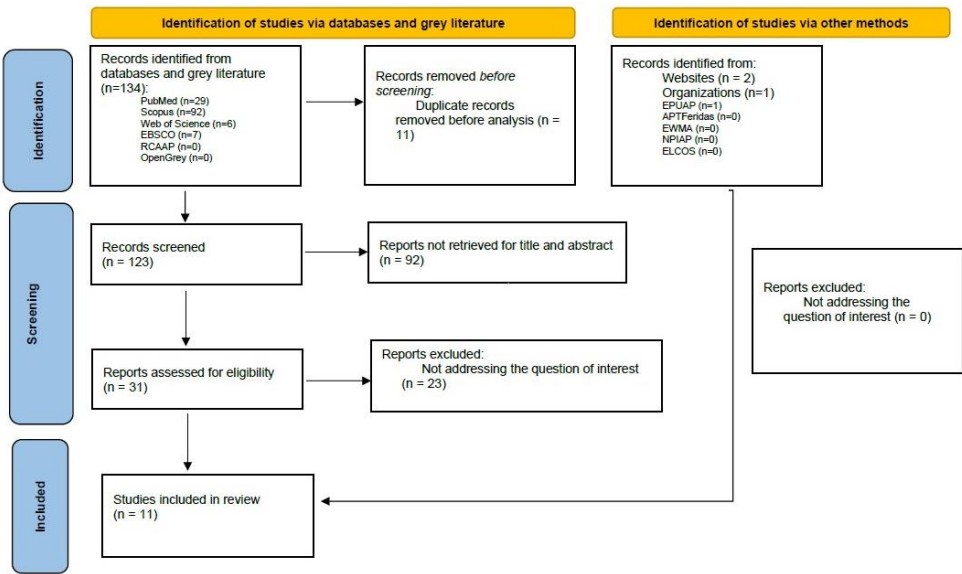

**Figure 1.** PRISMA diagram.

### 2.3. Ethical Considerations

Given the characterization of the study, in which no participants are included being a scoping review, there are no ethical implications to take into account.

### 3. Results

Of the 34 articles identified in this scoping review, 11 were included, with publication dates between 2015 and 2022: 9% (1) in 2015, 9% (1) in 2016, 9% (1) in 2017, 36% (4) in 2018, 9% (1) in 2019, 18% (2) in 2020 and 9% (1) in 2022.

Taking into account the type of study and the level of evidence, experimental/intervention studies were most commonly found (5), level of evidence III. Randomized controlled trials (2), level of evidence II, cohort studies (2), level of evidence (2), level of evidence IV and, finally, last, one review study (1), level of evidence VI, and a clinical guide based on systematic reviews or meta-analyses (1), level of evidence I, as shown in Table 2.

**Table 2.** Level of evidence from the studies.

| Level of Evidence | Studies | Frequency | Percentage |
|---|---|---|---|
| I | Systematic review and meta-analysis of randomized controlled trials; clinical guides based on systematic reviews or meta-analyses | 1 | 9% |
| II | Randomized controlled trials | 2 | 18% |
| III | Non-randomized clinical trial | 5 | 45% |
| IV | Case Studies and Cohort Studies | 2 | 18% |
| V | Systematic review of descriptive and qualitative studies | 0 | 0% |
| SAW | Descriptive or qualitative study | 1 | 9% |
| | Total | 11 | 100% |

Of the 11 articles included in the study, 3 are multicenter (27%) and the rest were developed in six different countries, namely: UK (27%), USA (9%), Japan (9%), Saudi Arabia (9%), Italy (9%), Spain (9%).

All analyzed articles refer to interventions for the prevention of skin lesions associated with the NIV mask. From these articles, data were extracted using a tool developed by the reviewers. This data extraction was performed by two independent reviewers and the relevant data included are presented in Table 3, which presents the interventions according to each article analyzed. Some of the articles refer to interventions that have an impact on nursing care.

**Table 3.** Articles included and relevant results.

| Title | Year | Intervention |
|---|---|---|
| *Pressure ulcer incidence in patients wearing nasal-oral versus full-face non-invasive ventilation masks* [8] | 2015 | • Evaluation of the skin every 12 h and whenever the mask is removed, <br> • Use of full-face mask |
| *Investigating the effects of strap tension during non-invasive ventilation mask application: A combined biomechanical and biomarker approach* [20] | 2016 | • Avoid an exaggerated strain on fixation as this leads to increased pressure and increased cytokines |
| *Preventing facial pressure ulcers in patients under non-invasive mechanical ventilation: A randomized control trial* [21] | 2017 | • Application of hyper-oxygenated fatty acids <br> • Skin evaluation and skin care under NIV mask every 4 to 6 h <br> • Correctly choose the size of the device and adjust its fixation |
| *Mask pressure effects on the nasal bridge during short-term non-invasive ventilation* [22] | 2018 | • Apply and adjust the NIV mask with the patient in dorsal decubitus |
| *Development of Personalized Fitting Device With 3-Dimensional Solution for Prevention of NIV Oronasal Mask-Related Pressure Ulcers* [13] | 2018 | • The use of a personalized adaptation device (to fill between the face and the mask), created with the aid of 3D scanning solutions |
| *Effect of humidified non-invasive ventilation on the development of facial skin breakdown* [23] | 2018 | • Have greater vigilance in the use of humidified CPAP, since this increases the risk of developing PU |
| *The Preventative Effect of Hydrocolloid Dressings on Nasal Bridge Pressure Ulceration in Acute Non-invasive Ventilation* [9] | 2018 | • Use of hydrocolloid plate under NIV mask |

**Table 3.** *Cont.*

| Title | Year | Intervention |
|---|---|---|
| *Prevention and Treatment of Injuries/Pressure Ulcers: Quick Reference Guide2019* [24] | 2019 | • Evaluate and select the device for: ability to minimize damage, shape and size appropriate to the individual, ability to use the device correctly according to the manufacturer's instructions and ability to fix the device correctly<br>• Regularly monitor the device for tension of the fixation and comfort of the person<br>• Frequently evaluate the skin under and around the device, looking for signs of pressure ulcer, as an integral part of skin evaluation<br>• Reduce or redistribute the pressure at the skin-device interface: regularly repositioning or rotating the medical device and/or the person, providing physical support to the device to prevent pressure and torsion, removing the device as early as the device is possible<br>• Use protective dressing (under the device) to aid in the prevention of MDRPUs |
| *Device-related pressure ulcers: SECURE prevention* [6] | 2020 | SPECIFIC INTERVENTIONS NIV:<br>• Put protective foam under all masks<br>• store protective dressings near masks and/or group them together; shape and adjust protective dressing using patient-specific models<br>• Do not wear poorly fitting whole face masks<br>• use custom mask adjustment devices, designed using three-dimensional scanning<br>• Use devices with surfaces appropriate to the size of the patient<br>• Assess the need for adhesives<br>• Inspect the skin for risk area and anatomical location, including face and scalp<br>• Rotate devices<br>• Optimize nutrition.<br>MENEMONIC SECURE:<br>• Skin evaluation<br>• Education of professionals, patients, caregivers and industry<br>• Adopt evidence-based devices<br>• Understand the causes of development of UPPADM and evaluate the correct use of the device<br>• Report the MDRPU<br>• Assess device capability and minimize the risk of MDRPUs |
| *Material and Technology: Back to the Future for the Choice of Interface for Non-invasive Ventilation—A Concise Review* [22] | 2020 | • The rotating use of different NIV interfaces |
| *Device-related pressure ulcers: Secure prevention* [25] | 2022 | • Consider switching interfaces to switch pressure points<br>• Watch the skin<br>• Use protective dressing (transparent film, hydrocolloids, foams, gel pads, silicone)<br>MENEMONIC SECURE:<br>• Skin evaluation<br>• Education of professionals, patients, caregivers and industry<br>• Adopt evidence-based devices<br>• Understand the causes of development of MDRPUs and evaluate the correct use of the device<br>• Report the MDRPU<br>• Assess device capability minimize the risk of MDRPUs |

Taking into account Table 3, several interventions are observed in the 11 studies. Some articles refer to specific interventions for the prevention of PU associated with the NIV mask, and others complement this, with general interventions for the prevention of these lesions. Interventions with different levels of complexity are presented, i.e., some studies indicate interventions such as skin inspection and, on the other hand, others indicate interventions that require specific equipment, such as personalized 3D devices. All of the interventions presented can bring benefits to the patient who needs NIV therapy. The education of professionals, patients themselves and their caregivers for the correct use of devices, as well as for their correct attachment, are also highlighted.

In Table 4 we present a summary of the interventions found in the included studies The most cited among these were the evaluation of the skin; however, studies indicate different time intervals for this inspection. The appropriate adjustment of the mask is important, with special attention to the tension of the fixing tapes. The application of protective dressing under the NIV mask is also an intervention proposed in several studies (for example, transparent film, hydrocolloids, foams, gel pads, silicone). The choice of the correct size of the interface, and if possible, use of a personalized device or 3D model, are also key. The alternation of the interface of NIV in order to alternate pressure points was also mentioned, since there are different masks that allow alternation of pressure points, such as the full-face mask alternating with the traditional mouth and nose mask.

**Table 4.** Summary of interventions found in the included studies.

| Intervention | Observation |
|---|---|
| Evaluation of the skin | Different ranges presented in the literature |
| The appropriate adjustment of the mask | Special attention of the fixing tapes |
| The choice of the correct size of the interface | If possible, use of a personalized adaptation device |
| The alternation of the interfaces of NIV | To alternate pressure points, for example use the full-face mask |
| The use of protective dressing under the NIV mask | transparent film, hydrocolloids, foams, gel pads, silicone |

## 4. Discussion

This scoping review was conducted to pinpoint interventions that mitigate skin lesions linked with NIV masks in adult patients. The review consolidates several studies, shedding light on interventions and recommendations that potentially elevate the care quality for patients utilizing non-invasive ventilation. For these patients, preventing these injuries can allow treatment to continue through NIV, and can make it more comfortable, allowing for a greater quality of life.

The identified interventions include skin assessment, optimal fixation of the device and the use of interfaces, e.g., transparent film, hydrocolloids, foams, gel pads, silicone dressings under the NIV mask, among others.

Skin inspection and evaluation was the most cited intervention in the 11 articles included in this review, however with different frequency among authors. In 2015, an author refers to in which circumstances skin should be evaluated every 12 h and whenever the NIV mask is removed [8]. In 2017, Otero et al. reiterates the importance of assessing the skin under the mask at intervals between 4 and 6 [21]. Other studies highlight the importance of evaluation and inspection of the skin under and around the NIV mask, without highlighting, however, an ideal periodicity [6,23,25]. Also noteworthy is the reference to a more rigorous monitoring of the skin when using humidified CPAP, as this increases the probability of developing PU [23], since a humidified NIV presents a potential adverse effect on the skin's barrier function, associated with changes in the microclimate. This has recently been recognized as a factor that reduces the skin's tolerance to pressure and shear, thus exposing it to an increased risk of development [23].

Aspects such as the selection of the ideal device, and the taking into account of the adequate size of the mask, its adjustment and correct fixation, with ideal tension in the fixation straps, avoiding exaggerated tension, are also interventions relevant to clinical practice, evidenced in several articles [6,21,25,26]. One study mentions, in particular, that the adjustment of the NIV mask must be performed with the patient in the supine position for an adequate positional and tension adjustment in the NIV mask fixation strips, as this allows greater distribution of pressure across the face, avoiding incorrect positioning of the device [22]. A 2016 study investigated the effects of belt tension on the application of NIV

masks, suggesting a biomechanical approach and the use of biomarkers, since these can help prevent skin lesions [20].

Brill et al., in 2018, [26] conducted a study that indicates that, for the prevention of PUs associated with the NIV mask, a personalized adaptation device (for filling in between the nose and the mask), created with the aid of 3D scanning solutions [13], should be used. This recommendation came to be reaffirmed in the International Consensus Document *"Device-related pressure ulcers: Secure prevention"*, updated in 2022 [25]. This solution would allow for a better individual adjustment of the device to the patient, avoiding the need for excessive tension.

Of the 11 articles included in this review, 4 consider the use of dressing material under the mask as relevant for the prevention of PU associated with the NIV mask. In 2018, a study demonstrated the preventive effect of hydrocolloid dressing on nasal bridge ulceration in patients undergoing acute NIV [9]. The EPUAP/NPIAP/PPPIA quick reference guide (2019) indicates the use of a protective dressing under the mask without specifying its composition [24]. In turn, the international consensus document SECURE (2020) recognizes the importance of using a foam pad under all masks, also warning of the need to store them near NIV masks [6]. In the update of the document previously mentioned in 2022, the range of options for transparent films, hydrocolloids, foams, gel pads, and silicone is extended, keeping the focus on the use of dressing material for the prevention of PU under the mask [25]. In 2017, a study indicated that the application of hyper-oxygenated fatty acids to the NIV mask pressure sites prevented PUs caused by them [21].

In the selected articles, we can also highlight other interventions to prevent PU associated with NIV masks, such as the rotational use of different NIV interfaces [22,23], the education of professionals and their understanding of the causes of MDRPU developments, the use of devices that have been developed based on scientific evidence and, finally, nutrition optimization, included in the recommendations of international consensus documents [6,23]. A 2015 study showed that the incidence of PU was lower in patients who used a full-face mask, when compared to patients who used a nasal-oral mask [8].

In summary, after analyzing the articles included in this scoping review, we can highlight a set of interventions to prevent PU associated with the NIV mask, frequently cited in several studies, such as skin inspection and assessment, although there is no evidence on the ideal periodicity of this intervention, the selection of the ideal device, taking into account its size and its characteristics, the ideal adjustment of the fixing strips, and the use of dressing material under the NIV mask to prevent PU associated with it, among others. We consider that, in clinical practice, depending on the context, bundles can be created to prevent PU associated with non-invasive masks, which are based on the interventions found in the literature, allowing their systematic application to all patients who require this medical device, aimed at preventing pressure ulcers. Furthermore, this topic can be addressed in training, so that all professionals are up-to-date and aware of complying with the same interventions. Therefore, by applying a multi-modal set of interventions, it may be possible to reduce the incidence rate of pressure ulcers associated with the NIV mask, as well as increase patient comfort during this therapy.

To encapsulate, this review foregrounds several interventions to preempt PU occurrence in conjunction with NIV masks. These interventions include periodic skin assessments (with no established ideal frequency), selecting and adjusting the device optimally, and employing specific dressing materials beneath the NIV mask.

## 5. Conclusions

There is still not enough evidence to support a decision on which appropriate approach is indicated for the prevention of skin lesions associated with the NIV mask. There is a multi-modal set of interventions in the literature, including skin assessment, material selection and proper adjustment of the mask, use of dressing material under the NIV mask, among others. This scoping review demonstrates that there is some clear scientific evidence, however the methodological approaches are very different, which makes it difficult to

clearly describe the referenced interventions. It is recommended that future investigations evaluate specific interventions; however, it would allow early intervention, becoming an important contribution to a better assessment of the person using non-invasive ventilation and, of course, obtaining more effective prevention rates, culminating in better care in this population and context.

There exists a pressing need for more rigorous experimental studies to shepherd health professionals towards evidence-based practices. In emphasizing this, we aspire to bolster the quality of care extended to individuals reliant on non-invasive ventilation.

In short, there is a set of interventions in the literature that allow a reduction in the incidence of pressure ulcers associated with IVN masks, including skin assessment, selection of the ideal device, and application of prophylactic dressings under the mask.

**Author Contributions:** Introduction, P.A. and R.A.; Materials and Methods, T.M.; Results, R.A. and T.M.; Discussion, R.A. and T.M.; Conclusion, R.A.; writing—original draft preparation, R.A.; writing—review and editing, T.M. and P.A. All authors have read and agreed to the published version of the manuscript.

**Funding:** This research received no external funding.

**Guidelines and Standards Statement:** This manuscript was drafted according to the recommendations of the Joanna Briggs Institute for scoping review research.

**Conflicts of Interest:** The authors declare no conflicts of interest.

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
