# Peer review of "Non-invasive Ventilation Interventions for Skin Injury Prevention: Scoping Review"

_nursrep, doi:10.3390/nursrep14010005_

Round 1

Reviewer 1 Report

Comments and Suggestions for Authors

Good scoping review

a table resuming the recommendations could be added

Author Response

Dear reviewer,

Thanks for your review. According to your recommendation, a table resuming the suggestions was added to the manuscript. Please check table 4.

Sincerely,

Reviewer 2 Report

Comments and Suggestions for Authors

Dear authors, 

Thank you for allowing me to evaluate and review your manuscript entitled "NONINVASIVE VENTILATION INTERVENTIONS FOR 2 SKIN INJURY PREVENTION: Scoping Review". This paper aimed to map the available literature and identify which interventions might help reduce pressure lectures in patients undergoing non-invasive ventilation.

I agree with the authors who decided to conduct a scoping review: I believe this is the correct methodological approach. In addition, an appropriate reporting guideline was used, respecting its structure.

The manuscript is generally well-written. However, I find the background inadequate for the context under study.

The important thing is to create a common thread between non-invasive ventilation —> skin injuries, —> interventions.

Your background introduced topics not subsequently covered in the text.

Currently, there is no critical appraisal for scoping review; for this reason, I used the “PRISMA scoping review” as an instrument of critical appraisal.

METHODOLOGY

Research Question: The research question is clearly stated and follows the PCC (Population, Concept, Context) framework, which is appropriate for a scoping review. The question pertains to interventions to prevent pressure ulcers associated with non-invasive ventilation masks in adults.

Inclusion Criteria: The inclusion criteria are well-defined and comprehensive. They specify the target population (adults aged 18 or older), the intervention (non-invasive ventilation masks to prevent pressure ulcers), and the wide range of study types to be considered. It's explicitly mentioned that health conditions and participant origin (home or hospitalisation) are not exclusion factors. This is important to ensure the inclusivity of the scoping review.

Research Strategy: The paragraph describes the research strategy, which involves searching multiple databases and including studies in English, Portuguese, and Spanish published from 2010 to the present. Please describe why all other languages were excluded (French, Deutsch, Sweden, etc.)

Additionally, the search extended to guidelines and protocols from relevant associations. This comprehensive approach is suitable for a scoping review.

Data Selection Process: The paragraph mentions that two independent reviewers selected articles based on titles and abstracts, which is a standard practice in systematic reviews. Removing duplicate studies is also noted, which is an essential step in the review process to ensure data accuracy.

Clarity and Organization: The paragraph is generally clear and well-organized, with distinct sections for methodology, inclusion criteria, research strategy, and data selection. However, there are some minor formatting issues, such as a missing space before "model" and a missing space before the hyphen in "non-invasive."

However, minor formatting issues should be addressed for clarity and completeness.

DISCUSSION

The provided discussion of the manuscript discusses the interventions identified in a scoping review focused on mitigating skin lesions associated with non-invasive ventilation (NIV) masks in adult patients. Below is a critical review of the discussion:

The discussion opens with a clear statement of the review's objective: pinpoint interventions to mitigate skin lesions linked with NIV masks.

In addition, it provides a comprehensive list of interventions identified in the scoping review, including skin assessment, device fixation, and the use of dressing materials. The information is well-organized and easy to follow.

Supporting Evidence: The discussion appropriately cites the studies and sources that support each intervention, enhancing the credibility of the findings. This is crucial for the reader to verify and understand the basis for the recommendations.

Practical Implications: The discussion briefly touches on the practical implications of the interventions, such as using 3D scanning for personalised adaptation devices. However, it could benefit from a more detailed exploration of how these interventions can be implemented in clinical practice.

Incomplete Citations: While the discussion mentions studies and authors (e.g., "Otero et al."), providing complete references or citations to these sources would be helpful. This ensures transparency and enables readers to locate the original studies for more in-depth exploration. Please insert the citation at the end of the following sentence: "In 2017, Otero et. al reiterates the importance of assessing the skin under the mask at intervals between 4 and 6 h."

Grammar and Typographical Errors: The discussion contains several typographical errors and grammatical issues. For example, "patientes" should be corrected to "patients," and there are missing spaces or punctuation issues in several places. Proper proofreading is necessary to improve the overall professionalism of the manuscript.

Conclusion and Recap: The conclusion neatly summarises the key interventions, emphasising skin assessment, device selection, and the use of dressing materials. However, the final sentence could be more concise, reiterating information already presented in the discussion.

In summary, the discussion provides a valuable overview of the identified interventions and their variations in recommendations. It correctly cites supporting sources but lacks complete references for some studies. Addressing the grammatical issues and expanding on the practical implications of the interventions would improve the overall quality of the discussion.

Author Response

Dear reviewer,

Thanks for your comments on our manuscript. We have taken into consideration all the comments made and tried our best to address them.

Regarding the background questions, we have clarified in the introduction section the relationship between “non-invasive ventilation —> skin injuries, —> interventions”. The highlighted text in the introduction page1 and 2 should demonstrate our effort.

Regarding your methodology analysis we have the following statements:

  • About the question in the research strategy, you can find our justification in page 4 line 134 to 137. The limitation of languages is due to native languages of the authors plus English to capture relevant international studies.
  • Under Clarity and Organization, we reviewed the typos identified and proof read our article once again to check others typos in the text. We believe we have addressed all.

Regarding your review of the discussion presented in our article:

  • Regarding practical implications we have further documented your analysis in page 10 line 267 to 274. We have contextualized how these intervention can be applied in clinical practice.
  • Regarding incomplete citations, we addressed your concerns and added the missing references as requested. Also, we have reviewed the references section for completion and correction of references 1, 18 and 21.
  • Regarding Grammar and Typographical Errors, we have identified and corrected all typos in the text. All changes are highlighted in the text and you can find all the corrections there.

Finally, addressing your comments on your conclusions:

  • We followed your suggestion and clarified our last sentence. We believe to have addressed the main issues raised by your review.

Overall, we believe that we improved the quality of the document with your and the other reviewer's comments and that the work we made in the 5 days granted to us shows our dedication.

Best regards,

Reviewer 3 Report

Comments and Suggestions for Authors

see file attached

Author Response

Dear reviewer,

Thanks for your comment on our work. We have revised the references section, including reference no18 as you pointed in your comments. Please check the document the changes made.

Sincerely,

Reviewer 4 Report

Comments and Suggestions for Authors

Minor changes:

Pag 2 line 73.  “use of protective apotheses” is not clear, consider to change to “protective dressings”. Also in table 3.  In other section of the manuscript ( line 192) authors use “prophylactic patch”. Please check, and use a correct name consistently.

Pg 2 line 98. Explain was type of study is “discretionary studies”.

Figure 1. Some parts of the text are in Portuguese not in English, please check

Check references 1 and 21 for correct citation.

Comments on the Quality of English Language

-

Author Response

Dear reviewer,

Thanks for your comments on our manuscript. We have revised the document taking in consideration your comments.

Regarding your comments about the expression protective apotheses, we have changed them to protective dressings throughout the document. So we have changed the expression both in pag.2 line72 and table 3.

In page 2 line 98 there was a typo in translation to English discretionary studies, so we have changed it into the correct term which is descriptive studies.

We have corrected Figure 1 and substituted the Portuguese text.

We have proof-read our work, correct all the mistakes enumerated in your comments and revised the references section including references no1 and 21, against the documents consulted as part of this scoping review. There were changes in their definition in the reference section.

Sincerely,

Reviewer 5 Report

Comments and Suggestions for Authors

scoping review is well designed, complies with all proposed methodological steps and the information is clear.

Well-qualified survey question.

I make small suggestions.

1) Page 1 line 29 - replace review UPPs. Wouldn't it be UP?

2) Table 1. Database search - the data bases:

Open Gray

EWMA

NPIAP

ELCOS

APT-Wounds

they do not have systematic data search strategies using descriptors. Do they need to be on this table?

Were the key search expressions using descriptors the same for all bases? PUBMED / SCOPUS / WEB OF SCIENCE?

Author Response

Dear reviewer,

Thank for your comments on our manuscript. We have revised the document according to your recommendations.

Regarding question 1, we changed the acronym UPPs to the correct term PU (pressure ulcer).

About question 2, we believe that table 1 has a meaningful value. Despite the fact that some databases do not support descriptors natively, we have conducted the same work, to the best of your effort manually. We have also used the same descriptors on all databases. You can check your clarification on this matter on page 4 lines 124 to 145.

Sincerely,

Reviewer 6 Report

Comments and Suggestions for Authors

Congratulations on the idea for a very interesting work, I read it carefully, I have comments on table 3, apart from the title and interventions, the authors' main conclusions should be taken into account and a citation should be included.

There are several incorrect quotations in the literature and the links are inactive, including: 21 should be there

Kottner J, Cuddigan J, Carville K, Balzer K, Berlowitz D, Law S, Litchford M, Mitchell P, Moore Z, Pittman J, Sigaudo-Roussel D, Yee CY, Haesler E. Prevention and treatment of pressure ulcers/injuries: The protocol for the second update of the international Clinical Practice Guideline 2019. J Tissue Viability. 2019;28(2):51-58

Author Response

Dear reviewer,

Thanks for your comments on our manuscript. We have revised the document according to your comments.

We have included citations in table 3 according to your comment.

We double checked the links in our references and corrected reference no 21 according to the document that was used. The correct citation as present in the document as suggestion to citation is: European Pressure Ulcer Advisory Panel, National Pressure Injury Advisory Panel and Pan Pacific Pressure Injury Alliance. Prevenção e tratamento de lesões / úlceras por pressão. Guía de consulta rápida. (edição Portuguesa). Emily Haesler (Ed.). EPUAP/NPIAP/PPPIA: 2019. We believe that this is the correct reference for the document.

Sincerely,